# A novel GTP-binding protein–adaptor protein complex responsible for export of Vangl2 from the *trans* Golgi network

Yusong Guo, Giulia Zanetti[†], Randy Schekman*

Department of Molecular and Cell Biology, Howard Hughes Medical Institute, University of California-Berkeley, Berkeley, United States

**Abstract** Planar cell polarity (PCP) requires the asymmetric sorting of distinct signaling receptors to distal and proximal surfaces of polarized epithelial cells. We have examined the transport of one PCP signaling protein, Vangl2, from the *trans* Golgi network (TGN) in mammalian cells. Using siRNA knockdown experiments, we find that the GTP-binding protein, Arfrp1, and the clathrin adaptor complex 1 (AP-1) are required for Vangl2 transport from the TGN. In contrast, TGN export of Frizzled 6, which localizes to the opposing epithelial surface from Vangl2, does not depend on Arfrp1 or AP-1. Mutagenesis studies identified a YYXXF sorting signal in the C-terminal cytosolic domain of Vangl2 that is required for Vangl2 traffic and interaction with the μ subunit of AP-1. We propose that Arfrp1 exposes a binding site on AP-1 that recognizes the Vangl2 sorting motif for capture into a transport vesicle destined for the proximal surface of a polarized epithelial cell.

*For correspondence:
schekman@berkeley.edu

†Present address: School of Crystallography, Birkbeck College, London, United Kingdom

## Introduction

Planar cell polarity (PCP) governs the organization of epithelial cells along a plane parallel to the surface of the epithelium. This long range order orchestrates proper development and organ function. The establishment of PCP is regulated by a set of evolutionarily conserved signaling receptors. A key feature of these signaling receptors is that they are asymmetrically localized on the cell boundaries during PCP signaling (*Klein and Mlodzik, 2005*). The mechanisms that mediate the asymmetric localization of PCP signaling molecules remain unclear. One hypothesis is that interactions between PCP signaling molecules across cell junctions could stabilize their polarized localization to opposing cell boundaries (*Klein and Mlodzik, 2005*; *Chen et al., 2008*). Proteins that organize epithelial cells include the atypical cadherin Fat, Dachsous and the Golgi resident protein Four-jointed in *Drosophila* which have been proposed to provide long range patterning cues to regulate PCP asymmetry (*Bayly and Axelrod, 2011*). Additional evidence suggests that intracellular trafficking may also contribute to the asymmetric localization of PCP signaling receptors (*Shimada et al., 2006*; *Strutt and Strutt, 2008*).

Coat-protein-mediated cargo protein sorting at the *trans* Golgi network (TGN) is an essential step of biosynthetic trafficking and regulates targeting of a variety of transmembrane cargoes to their final destinations (*Rodriguez-Boulan et al., 2005*). Among the known vesicle coat proteins, clathrin adaptor complexes (AP) have been shown to mediate sorting of various transmembrane cargoes at the TGN by directly interacting with tyrosine- or dileucine-based sorting motifs localized within the cytosolic domain of a transmembrane cargo molecule (*Rodriguez-Boulan et al., 2005*; *Burgos et al., 2010*). Recently, AP-1 has been shown to functionally interact with a novel Golgi-export motif within the tertiary structure of Kir2.1 channel (*Ma et al., 2011*). In addition to APs, a new type of coat protein complex, exomer, regulates the transport of Chs3p and Fus1p from the TGN to the plasma membrane

**eLife digest** Most cells in multicellular organisms possess a property known as polarity that is reflected, in part, in the organization of the cell surface into distinct domains. One well-known axis in epithelial cells, such as those in the skin, divides the cell into an apical domain, which faces out, and a basal domain, which faces the underlying tissue. These cells rely on the distribution of structural components inside the cell, or within the cell membrane, to tell the difference between these two directions. Epithelial cells also possess a second type of polarity, planar cell polarity, that ensures that cells adjacent to each other in the plane parallel to the skin tissue are oriented correctly with respect to each other during development. This ensures, in turn, that hairs, scales, feathers and so on are all aligned.

All eukaryotic cells sort and process proteins within an organelle called the Golgi apparatus, and proteins that are required at a specific destination within the cell, such as the cell surface membrane, carry specific molecular sorting signals that act as address labels to convey the protein into and within the secretory pathway. As one of these proteins moves through the Golgi apparatus, its sorting signals are recognized by coat proteins, such as clathrin, that subsequently form a vesicle around it. The assembly of this vesicle is initiated by an enzyme from the Arf family, but the enzyme must first undergo a conformational change (by exchanging a molecule of GDP for one of GTP) before formation can begin. The resulting vesicle can then be sent on its way to the address indicated by its Golgi-to-cell-surface sorting signal. These sorting signals also help to establish planar cell polarity in cells by ensuring that proteins called signaling receptors are distributed asymmetrically within the cell membrane.

Guo et al. have now examined the mechanism behind the asymmetric sorting of two proteins that are involved in planar cell polarity: Vangl2 and Frizzled 6. In an effort to understand why these proteins are localized to opposite surfaces of epithelial cells, Guo et al. used genetic techniques to reduce the expression of Golgi-localized Arf proteins in epithelial cell cultures. They found that knockdown of a protein called Arfrp1 caused Vangl2 to accumulate in the last station of the Golgi complex instead of being transported to the cell surface membrane. Then, using a technique called affinity chromatography, they demonstrated that a coat protein called the clathrin adaptor complex (AP-1) had to be present for the formation of vesicles around Vangl2. Moreover, disrupting AP-1 and Arfrp1 did not prevent Frizzled 6 being transported to the cell surface membrane. This suggests that cells use several distinct adaptor proteins and coat complexes to ensure that proteins from the Golgi apparatus go to specific locations on the cell surface and, thus, help to establish planar cell polarity.

in yeast (*Wang et al., 2006*; *Barfield et al., 2009*). Sorting of some soluble secretory cargo at the TGN requires the actin-severing protein ADF/cofilin and the $Ca^{2+}$ATPase SPCA1 (*von Blume et al., 2009*, *2011*; *Curwin et al., 2012*).

Assembly of coat protein complexes on membranes is initiated by Arf or Arf-like small GTPases that switch between GDP- and GTP-bound states. Upon GTP binding, Arf proteins expose an N-terminal myristoyl group attached to an amphipathic helix which mediates membrane recruitment and induces membrane curvature (*Lee et al., 2004*, *2005*; *Bielli et al., 2005*; *Beck et al., 2008*). GTP-binding also causes a conformational change in the switch domain of Arf proteins which promotes the membrane recruitment of cytosolic effectors, including coat proteins and lipid modification enzymes (*Gillingham and Munro, 2007*; *Donaldson and Jackson, 2011*). Mammalian cells possess 6 Arf proteins and more than 20 Arf-like proteins. The intracellular roles of the majority of Arf proteins are poorly understood. A genome-wide RNA interference screen indicates that Arf1 and Arfrp1 are required for secretion of recombinant luciferase from *Drosophila* S2 cells (*Wendler et al., 2010*). Arf1 regulates the membrane recruitment of various proteins including coats such as COPI, APs, GGAs and the lipid modification enzymes, phospholipase D and PtdIns 4-kinase (*Donaldson and Jackson, 2011*). Arfrp1 is essential for survival and has been shown to mediate the trafficking of VSVG, E-cadherin and the glucose transporters GLUT4 and GLUT2 as well as to regulate lipid droplet growth (*Shin et al., 2005*; *Zahn et al., 2008*; *Nishimoto-Morita et al., 2009*; *Hesse et al., 2010*; *Hommel et al., 2010*; *Hesse et al., 2012*) but the molecular mechanisms underlying its intracellular function are unknown.

Given the asymmetric distribution of PCP signaling molecules on the surface of epithelial cells, distinct sorting or coat protein complexes may be required for their traffic from the TGN. In this study, we focused on identifying the coat proteins that mediate TGN export of a conserved four-transmembrane PCP signaling receptor, Vangl2. In *Drosophila*, mutation in *Strabismus*, the *Drosophila* homolog of Vangl2, causes defects in the organization of wing hairs and induces defects in the orientation of eye ommatidia (*Taylor et al., 1998*; *Wolff and Rubin, 1998*). In vertebrates, Vangl2 regulates convergent extension (*Torban et al., 2004*). Mouse Vangl2 looptail mutants, which are defective in ER export, cause severe defects in neural tube closure and disrupt the orientation of stereociliary bundles in mouse cochlea (*Kibar et al., 2001a*, *2001b*; *Montcouquiol et al., 2003*; *Merte et al., 2010*).

To explore the coat proteins that mediate TGN export of Vangl2, we started by screening the effects on Vangl2 trafficking upon siRNA knockdown of selected Golgi-localized Arf proteins. Our analysis indicates that Arfrp1 regulates TGN export of Vangl2. We find that AP-1 is an effector of Arfrp1 and that the two interact to regulate TGN export of Vangl2. Interestingly, TGN export of one other PCP signaling receptor, Frizzled-6, is independent of the Arfrp1/AP-1 machinery, suggesting that differential sorting machineries regulate the TGN export of Vangl2 and Frizzled 6, which may contribute to their opposing localization on the epithelial cell surface.

## Results

### Knockdown of Arfrp1 accumulates Vangl2 at the TGN

To identify the Arf proteins that regulate TGN export of Vangl2, we performed an siRNA knockdown screen focusing on selected Golgi-localized Arf proteins in HeLa cells stably expressing HA-Vangl2. The screen indicated that knockdown of Arf1 or Arfrp1 caused a juxtanuclear accumulation of Vangl2 whereas knockdown of other Golgi-localized Arfs did not affect Vangl2 trafficking. Arf1, which shares a 34% sequence identity with Arfrp1, plays a general role in regulating membrane recruitment of various vesicle coat proteins and lipid modification enzymes (*Donaldson and Jackson, 2011*). Arfrp1 is more specifically localized at the TGN and has been shown to regulate TGN-to-plasma membrane transport of E-cadherin and VSV-G (*Shin et al., 2005*; *Zahn et al., 2008*; *Nishimoto-Morita et al., 2009*). However, what Arfrp1 does to promote traffic has not been explored. We thus focused on Arfrp1 and it's role in the transport of PCP signaling proteins. The expression of Arfrp1 was efficiently reduced after siRNA treatment (*Figure 1G*) and knockdown of Arfrp1 caused a juxtanuclear accumulation of Vangl2 in a majority (65%) of the cells compared to mock treated cells (*Figure 1A,D,H*). Transport-arrested Vangl2 colocalized with the TGN marker, Golgin 97 (*Figure 1A–F*) but not the early endosomal marker EEA1, the late endosomal marker Rab7 or the recycling endosomal marker Rab11 (*Figure 1—figure supplement 1*). Quantification of colocalization indicated that Vangl2 correlated more closely with the TGN marker, Golgin 97, than with the *cis*-Golgi marker, GM130 (*Figure 1—figure supplement 2*). These results suggest that Arfrp1 regulates the export of Vangl2 from the TGN.

### Subunits of the adaptor complex-1 preferentially bind the GTP-bound Arfrp1

To elucidate the roles of Arfrp1 in TGN export of Vangl2, we sought to identify the effectors of Arfrp1 using affinity chromatography. A similar approach documented the specific interaction between the BBsome, which functions as a coat complex that sorts membrane proteins to primary cilia, and Arl6 (*Jin et al., 2010*). Bovine brain cytosol was incubated with purified GST-tagged Arfrp1 dominant negative (T31N) and dominant active (Q79L) mutant pre-loaded with GDP or GTPγS, respectively. After incubation, bound proteins were eluted and analyzed by SDS-PAGE and silver staining. A series of protein bands were recovered in the eluate of GTPγS-loaded GST-Arfrp1 (Q79L) immobilized on glutathione beads (*Figure 2A*). One of the bands was identified by mass spectrometry as the γ subunit of the adaptor complex 1 (AP-1) (*Figure 2A*). Immunoblot analysis confirmed that both γ1-adaptin and μ1-adaptin preferentially interacted with the GTPγS-loaded Arfrp1 (Q79L), whereas EEA1, CRMP2 and dynamin II showed no binding or no GTP-dependent binding (*Figure 2B,C*). Moreover, the δ subunit of AP-3 and the α subunit of AP-2 showed no detectable binding (*Figure 2C*), suggesting the interactions between Arfrp1 and subunits of AP-1 are specific.

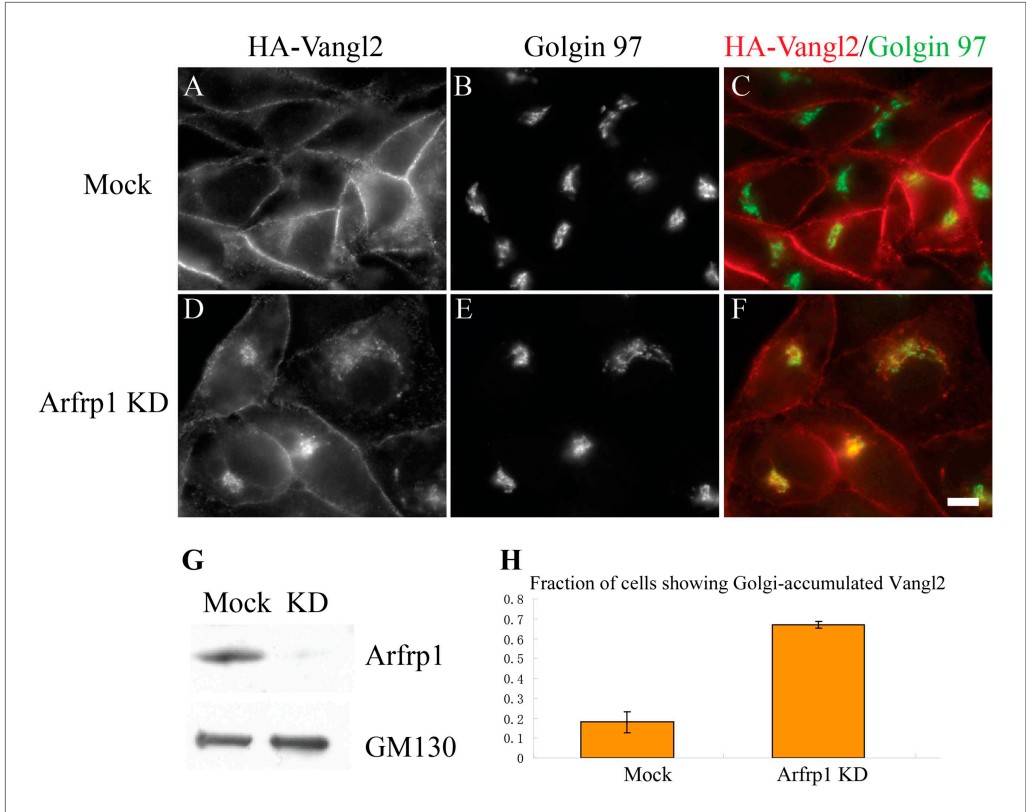

**Figure 1**. Knockdown of Arfrp1 leads to accumulation of Vangl2 at the TGN. (**A**)–(**F**) HeLa cells stably expressing HA-Vangl2 were either mock transfected or transfected with siRNA against Arfrp1. At day 3 after transfection, the cells were analyzed by indirect immunofluorescence. Size bar = 10 μM. (**G**) HeLa cell lysates from cells transfected with control siRNA or siRNA against Arfrp1 were analyzed by immunoblotting with anti-Arfrp1 antibody and, as a loading control, anti-GM130 antibody. (**H**) Quantification of the fraction of cells showing Golgi-accumulated Vangl2 in control or siRNA-treated HeLa cells stably expressing HA-Vangl2 (N = 3; >100 cells counted for each experiment).

The following figure supplements are available for figure 1:

**Figure supplement 1**. Juxtanuclear accumulated Vangl2 in Arfrp1 knockdown cells is not colocalized with endosomal markers.

**Figure supplement 2**. Juxtanuclear accumulated Vangl2 in Arfrp1 knockdown cells colocalizes with Golgin 97 more than with GM130.

## TGN export of Vangl2 depends on the conserved YYXXF motif at the C-terminal cytosolic domain

The results from the affinity isolation suggest that AP-1 is an effector of Arfrp1, possibly cooperating to mediate TGN export of Vangl2. Consistent with this hypothesis, the C-terminal cytosolic domain of Vangl2 contains a conserved basolateral-sorting motif (YXXF) which is known to interact with the AP complexes (*Bonifacino and Lippincott-Schwartz, 2003*) (red box, *Figure 3A*). Indeed, HA-Vangl2 is localized basolaterally in MDCK cells (*Kallay et al., 2006*). To test whether this motif is important for the localization of Vangl2, we generated a series of HA-Vangl2 mutant constructs and examined their localization. Strikingly, four Vangl2 mutants bearing mutations in the YXXF motif, including the single mutation (F283A), showed no detectable surface pattern (*Figure 3E,H,K,Q*). At high levels of expression, mutant Vangl2 was retained in the ER. However, at lower levels of expression, these Vangl2 mutant proteins accumulated in the juxtanuclear area which colocalized with the TGN marker, Golgin 97 (*Figure 3E–M,Q–S*). A Vangl2 YXXF double mutant (Y280A, F283A) and Vangl2 looptail mutant (D255E) displayed quite distinctive localization to the TGN and ER, respectively (*Figure 3—figure*

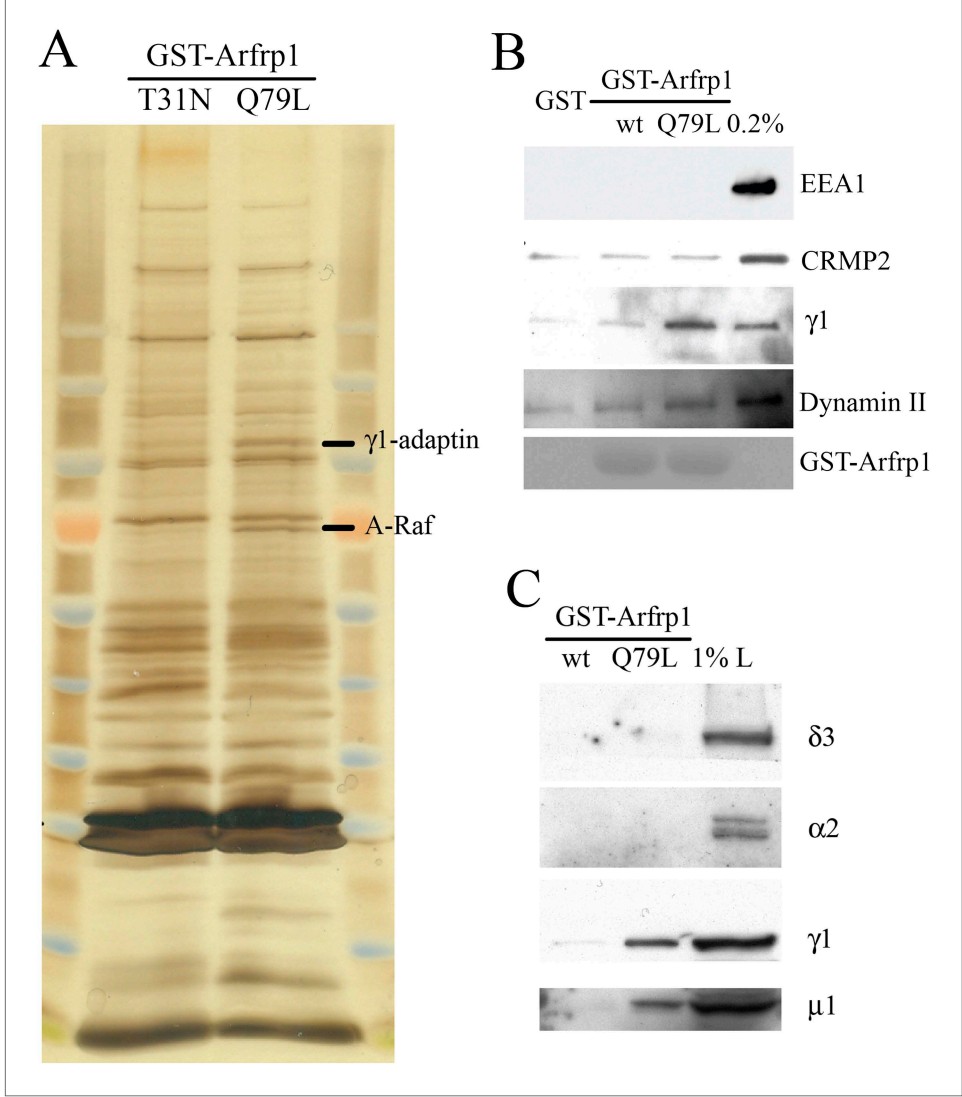

**Figure 2**. Subunits of AP-1 preferentially interact with the GTP-bound Arfrp1. (**A**) Bovine brain cytosol was incubated with purified GDP-loaded dominant negative form (T31N) or GTPγS-loaded dominant active form (Q79L) of GST-Arfrp1. After incubation, the eluted fraction was resolved by SDS-PAGE and silver stained. Protein identification in the indicated gel slice performed by mass spectrometry revealed γ1-adaptin and serine/threonine-protein kinase (A-Raf) respectively. (**B**),(**C**). Bovine brain cytosol was incubated with purified GDP-loaded GST-Arfrp1 (wt) or GTPγS-loaded GST-Arfrp1 (Q79L). After incubation, the entire sample of bound γ1-adaptin, μ1-adaptin and other indicated proteins was analyzed by immunoblot.

supplement 1). A single tyrosine mutant, Vangl2 Y280A, was only partially transport defective (*Figure 3N–P*), whereas the double mutant Y279A Y280A resulted in a more complete arrest of mutant Vangl2 at the TGN (*Figure 3—figure supplement 2*). As a control, substituting alanine for both leucines adjacent to the YXXF motif (green box, *Figure 3A*) had no effect on Vangl2 localization (*Figure 3T–V*). These results suggest that TGN export of Vangl2 depends on the conserved YYXXF motif in the C-terminal, cytosolic domain.

## μ1-adaptin directly interacts with Vangl2 in an YYXXF motif-dependent manner

The tyrosine-based sorting motif is known to interact with the μ subunit of the AP complexes (*Bonifacino and Lippincott-Schwartz, 2003*). To test whether μ1-adaptin interacts with Vangl2 via the YYXXF motif, we performed GST pull-down assays using purified GST-μ1 and lysates from COS7 cells

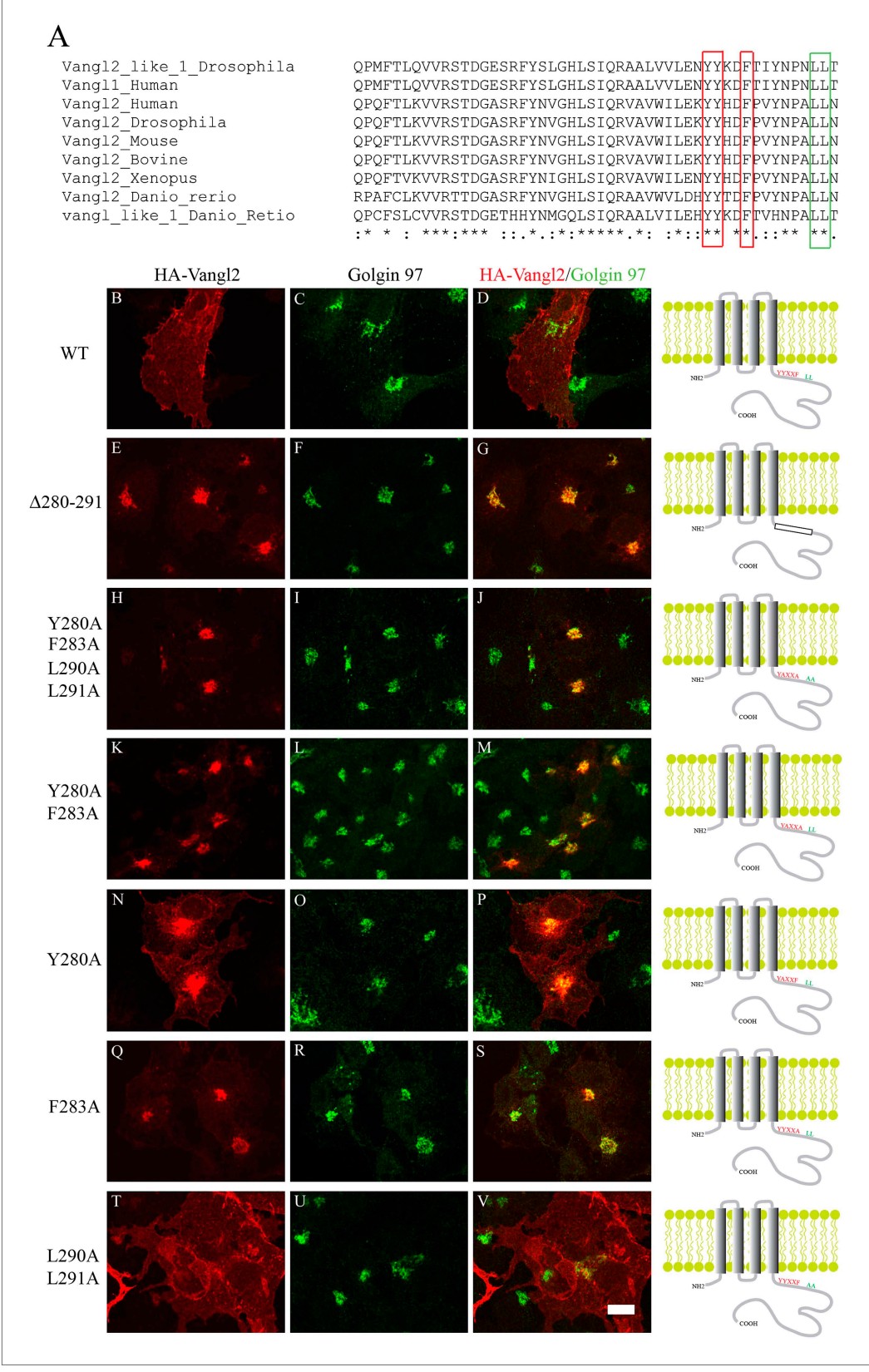

**Figure 3**. TGN export of Vangl2 depends on the conserved YYXXF sorting motif in the C-terminal cytosolic domain. (**A**) Sequence alignment of Vangl1 and Vangl2 from different species indicates that Vangl2 C-terminal cytosolic domain contains a conserved YYXXF sorting motif. (**B**)–(**V**) COS7 cells were transiently transfected with
*Figure 3. Continued on next page*

*Figure 3. Continued*
plasmids encoding HA-Vangl2 wild type (**B**–**D**) or the indicated mutant constructs (**E**–**V**). At day 1 after transfection, the cells were analyzed by indirect immunofluorescence using antibodies against HA tag and Golgin 97. Note the contrast in panel N was adjusted to reveal the weak surface pattern of Vangl2. Size Bar = 10 µM.
The following figure supplements are available for figure 3:
**Figure supplement 1**. Vangl2 tyrosine mutants are not colocalized with the ER marker.
**Figure supplement 2**. Vangl2 Y279A Y280A is blocked at the TGN.

transiently transfected with HA-Vangl2 wild-type or mutant constructs. HA-Vangl2 wild type specifically bound GST-µ1 (*Figure 4A*). The interaction between Vangl2 and GST-µ1 was severely reduced when crucial residues of the YYXXF motif were mutated, whereas alanine substitutions of the adjacent dileucine amino acids had no effect (*Figure 4A*). Yeast two-hybrid analysis confirmed that µ1-adaptin interacted with Vangl2 and mutation of the basolateral sorting motif, including the restrictive F283A substitution, inhibited this interaction (*Figure 4B*). The less restrictive single tyrosine mutant, Vangl2 (Y280A), interacted weakly with µ1-adaptin whereas mutating both tyrosine residues blocked interaction (*Figure 4B*). To test whether the Vangl2 cytosolic domain directly interacts with µ1-adaptin, we purified GST-µ1 and MBP-tagged Vangl2 C-terminal domain proteins. The MBP-Vangl2 C-terminal domain bound GST-µ1 whereas mutation of the YYXXF motif blocked this interaction (*Figure 4C*), consistent with a direct and signal-dependent interaction. The interaction pattern correlated well with the Vangl2 mutant localization analysis in transfected cells (*Figure 3B–V*).

## Knockdown of µ1-adaptin or γ1-adaptin accumulates Vangl2 at the TGN

To test whether AP-1 mediates TGN export of Vangl2, we knocked down the expression of the µ and γ subunits of AP-1 in HeLa cells transiently transfected with HA-Vangl2. Immunoblot analysis showed that the expression of µ1- and γ1-adaptins was significantly reduced after siRNA treatment (*Figure 5A*). As before, we focused on cells expressing lower levels of HA-Vangl2 and observed an accumulation of HA-Vangl2 in the juxtanuclear area, colocalized with Golgin 97, with weak or no detectable surface labeling in over 60% of the treated cells (*Figure 5E–J* and quantification in *Figure 5K*). Around 20% of mock-treated cells displayed Golgi-localized Vangl2 (*Figure 5K*) but retained strong surface labeling. As a control, knockdown of µ3-adaptin, which did not bind Vangl2 (not shown), or knockdown of δ3-adaptin had no significant effects on the localization of Vangl2 (*Figure 5K*). The interaction data and knockdown analysis suggest that AP-1 directly mediates TGN export of Vangl2.

## Interaction between Arfrp1, AP-1 and the Vangl2 cytosolic domain on synthetic liposomes

In order to assess the role of Arfrp1 and AP-1 in the sorting of Vangl2, we evaluated the interaction of pure components with synthetic membranes. First, we examined the recruitment of AP-1 to activated Arfrp1 using a liposome flotation assay. Purified Arfrp1-His associated with liposomes in the presence of GTPγS but not GDP (*Figure 6A*). Using the same flotation assay, we observed AP-1 complex recruited to liposomes incubated with GTPγS-Arfrp1-His, but not to those incubated with GDP-Arfrp1-His (*Figure 6B,C*). These results suggest that Arfrp1 binds AP-1 on the surface of liposomes in a concentration-dependent manner.

Next, we sought to analyze whether Arfrp1, in association with AP-1, could recruit Vangl2 cytosolic domain to liposomes. We were unable to address the recruitment of Vangl2 directly because purified Vangl2 cytosolic domain bound liposomes by itself. As an alternative approach, we evaluated the influence of the Vangl2 cytosolic domain and Arfrp1-GTPγS on the membrane recruitment of AP-1. As shown in *Figure 6D,E*, membrane recruitment of AP-1 was enhanced approximately threefold in the presence of both Vangl2 cytosolic domain and Arfrp1-GTPγS. Importantly, a Vangl2 sorting signal mutant, Y279A Y280A, failed to stimulate Arfrp1-mediated AP-1 recruitment. These results suggest that the Vangl2 sorting signal enhances AP-1 recruitment to membranes containing Arfrp1-GTP.

Arf1 also mediates membrane recruitment of AP-1. A peptide containing the mannose-6-phosphate receptor sorting signal stimulates Arf1-mediated membrane recruitment of AP-1 to liposomes (*Zhu et al., 1998*, *1999*; *Lee et al., 2008*). We evaluated the effect of the Vangl2 cytosolic domain on Arf1-mediated AP-1 recruitment using FLAG-tagged Arf1 and Arfrp1 purified from mammalian cells. In

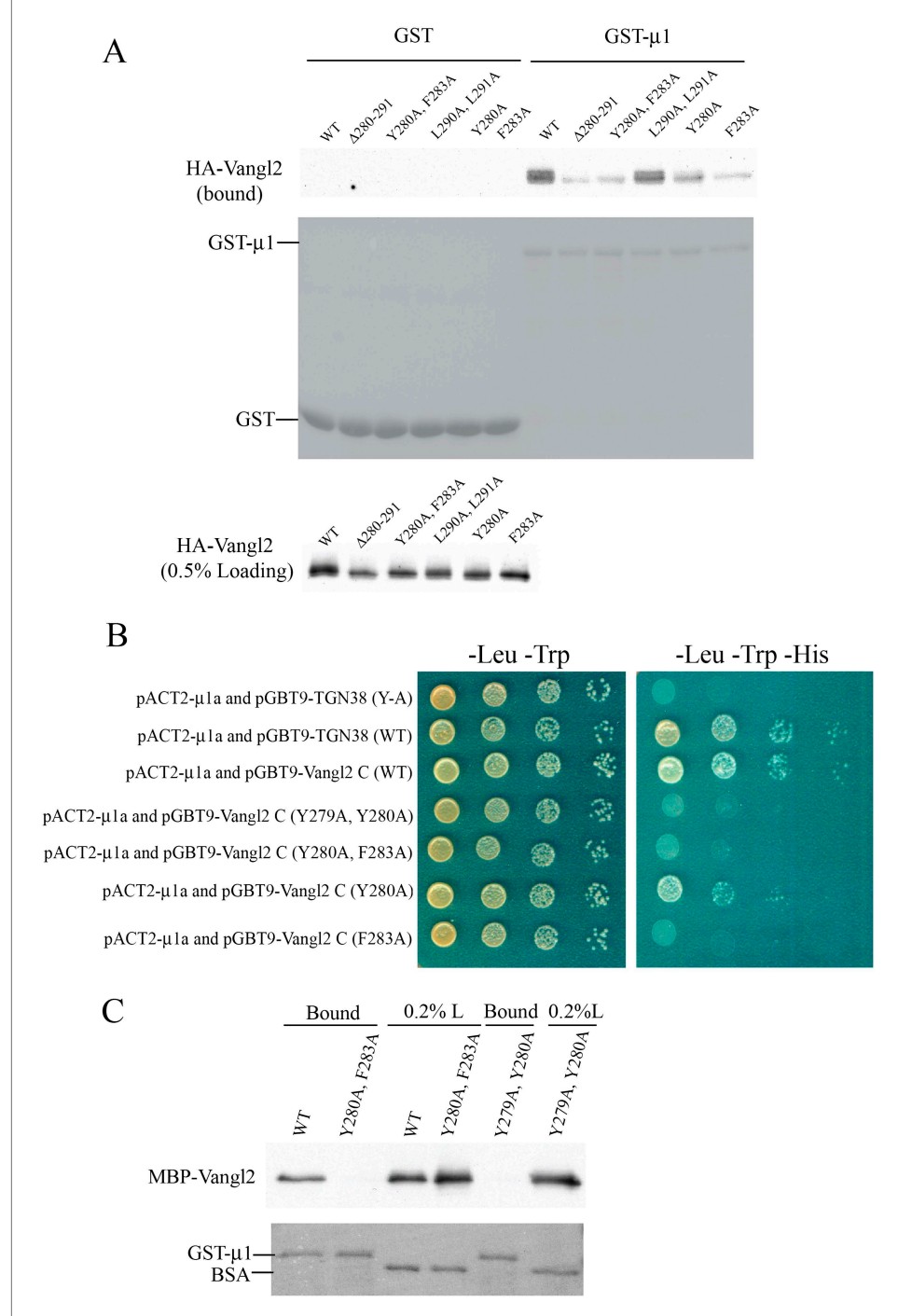

**Figure 4**. µ1-adaptin directly interacts with Vangl2 C-terminal cytosolic domain in an YYXXF-motif dependent manner. (**A**) Cell lysates from COS7 cells transiently transfected with plasmids encoding HA-Vangl2 wild-type or the indicated Vangl2 mutant constructs were incubated with glutathione beads bearing similar amounts of GST or GST-µ1. The entire sample of bound HA-Vangl2 was evaluated by immunoblotting with anti-HA antibody. (**B**) Yeast two-hybrid analyses recapitulated the results of the GST-pull down assay. Serial dilutions of the yeast colonies co-expressing the indicated constructs were dotted on the correspondent selective media. Pictures were taken after 3 days of growth. (**C**) Purified MBP-Vangl2 C-terminus wild type, or the indicated mutant constructs were incubated with glutathione beads bearing GST-µ1. The entire sample of bound MBP-Vangl2 was evaluated by immunoblot.

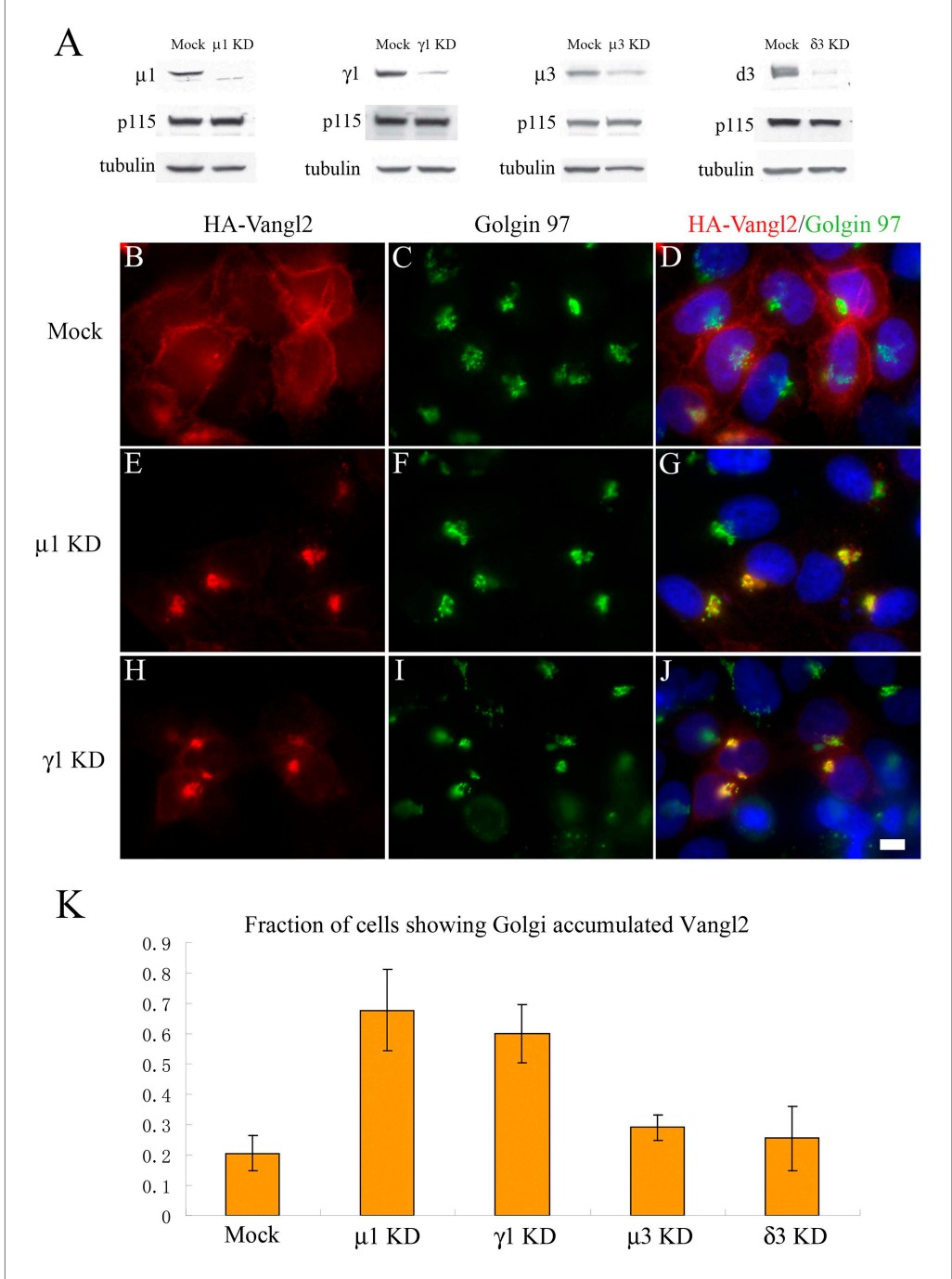

**Figure 5**. Knockdown of μ1-adaptin or γ1-adaptin accumulates Vangl2 at the TGN. (**A**) HeLa cells were mock transfected or transfected with siRNA against the indicated subunit of the AP-1 or AP-3 complex. At day 3 after transfection, total cell lysates were analyzed by immunoblotting with antibody against the indicated adaptin subunits or, as loading controls, p115 and tubulin. (**B**)–(**J**) HeLa cells were mock transfected (**B**–**D**) or transfected with siRNAs against μ1-adaptin (**E**–**G**) or γ1-adaptin (**H**–**J**) and re-transfected after 48 hr with plasmid encoding HA-Vangl2. After an additional 24 hr, cells were analyzed by immunofluorescence. Size bar = 10 μM. (**K**) Quantification of the fraction of cells showing Golgi-accumulated Vangl2 (N = 3; >150 cells expressing lower levels of Vangl2 counted for each experiment).

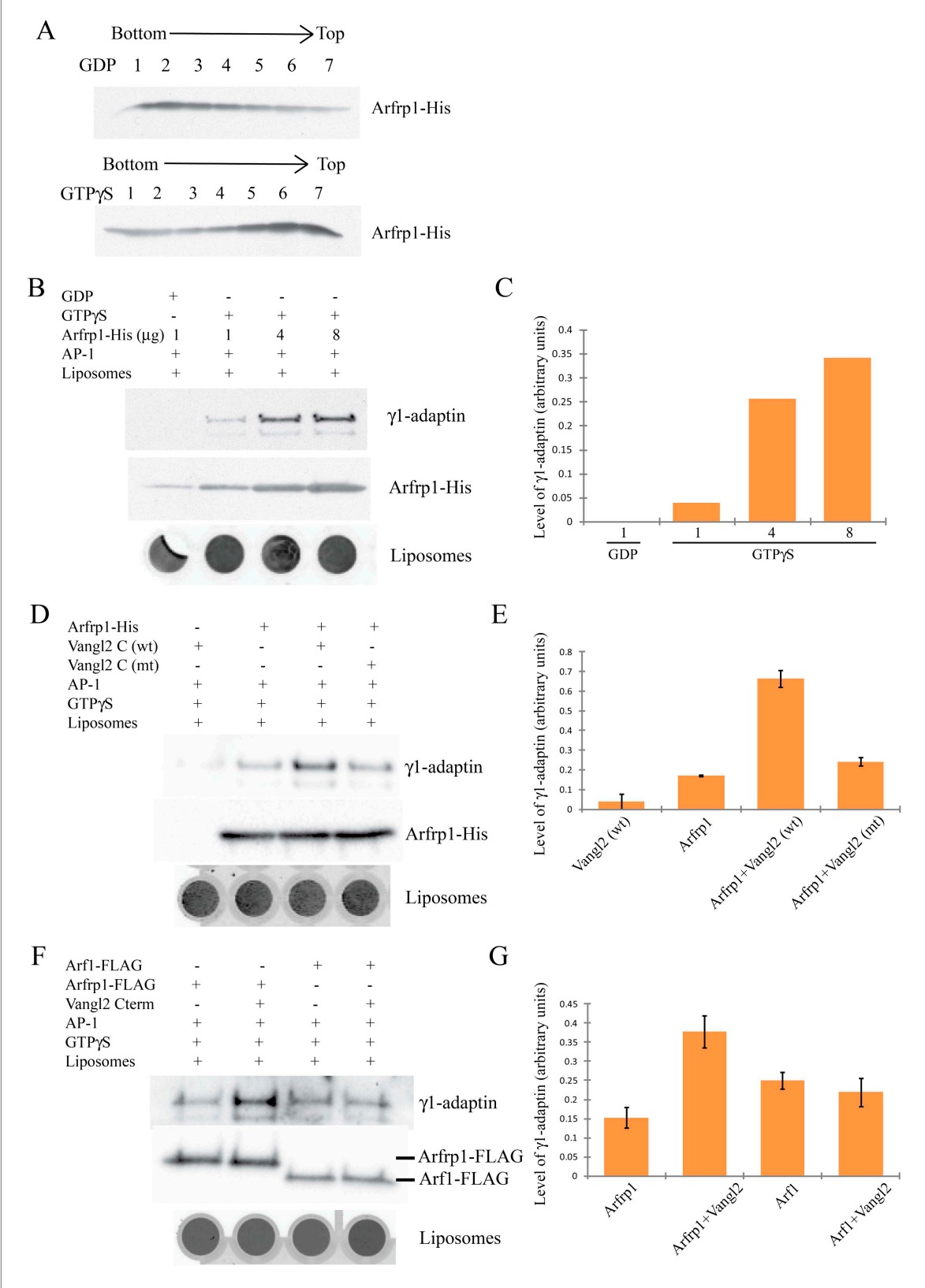

**Figure 6**. Arfrp1 directly recruits purified AP-1 complex to liposomes and this process is stimulated by Vangl2 C-terminal cytosolic domain. (**A**) Purified Arfrp1-His was incubated with liposomes labeled with Texas Red-PE in the presence of GDP or GTPγS. After centrifugation, fractions were collected from the bottom to the top and analyzed by immunoblotting using anti-His antibody. (**B**),(**C**). Liposomes were sequentially incubated with Arfrp1-His at the

*Figure 6. Continued on next page*

*Figure 6. Continued*
indicated concentration in the presence of GDP or GTPγS, then with purified AP-1 complex. After centrifugation, the top fractions were collected, scanned to reveal fluorescence in the Texas Red channel as an indicator of the amount of liposomes and analyzed by immunoblotting using anti-His and anti-γ1 antibodies (**B**) and the levels of γ1-adaptin normalized to the amount of lipids were quantified (**C**). (**D**),(**E**). Liposomes were sequentially incubated with Arfrp1-His alone or Vangl2 cytosolic domain alone or both, then with purified AP-1 complex. After centrifugation, the top fractions were collected, scanned to reveal fluorescence in the Texas Red channel, and analyzed by immunoblotting using anti-γ1 and anti-His antibodies (**D**) and the levels of γ1-adaptin normalized to the amount of lipids were quantified (**E**, N =2). (**F**),(**G**). Liposomes were sequentially incubated with Arfrp1-FLAG or Arf1-FLAG in the presence or absence of Vangl2 cytosolic domain, then with purified AP-1 complex. After centrifugation, the top fractions were collected, scanned to reveal fluorescence in the Texas Red channel and analyzed by immunoblotting using anti-γ1 and anti-FLAG antibodies (**F**) and the levels of γ1-adaptin normalized to the amount of lipids were quantified (**G**, N = 3).
The following figure supplements are available for figure 6:

**Figure supplement 1**. Sorting signal-dependent binding of Arfrp1 to Vangl2 in cell lysates; Vangl2 binds Arfrp1 more efficiently than Arf1.

contrast to incubations containing Arfrp1-GTPγS, Vangl2 C-terminal domain did not stimulate AP-1 recruitment to liposomes in the presence of Arf1-GTPγS (*Figure 6F,G*). This result suggests that the stimulation effect is specific for Arfrp1 and indicates that Arfrp1- but not Arf1- associated AP-1 provides a preferred binding site for the Vangl2 sorting signal. As expected, HA-Vangl2 from COS7 cell lysates interacted with GST-Arfrp1 but weakly with GST-Arf1 (*Figure 6—figure supplement 1A*). The interaction between GST-Arfrp1 and HA-Vangl2 depended on the YYXXF motif (*Figure 6—figure supplement 1B*) suggesting that Arfrp1 interacts with Vangl2 indirectly through the AP-1 complex.

## TGN export of two other PCP signaling receptors, Frizzled-6 and Celsr1, is independent of the Arfrp1/AP-1 machinery

Vangl2 and Frizzled-6 localize on opposing surfaces at cell–cell junctions in epithelial tissues. Because the TGN is a cargo sorting station, it is possible that Frizzled-6 and Vangl2 may use different vesicle sorting machineries to exit the TGN. Unlike Vangl2, Frizzled 6 was inefficiently transported to the cell surface in transfected HeLa cells. However, when Frizzled-6 was co-expressed with Celsr1, an atypical cadherin, both proteins co-localized at cell junctions (*Figure 7A–C*) (*Devenport and Fuchs, 2008*). Unlike Vangl2, knockdown of Arfrp1 or μ1-adaptin had no detectable effects on the localization of Frizzled-6 and Celsr1 (*Figure 7D–I*). Frizzled-6 and Celsr1 have no known tyrosine- or dileucine-based sorting motifs in their cytosolic domains. To test whether Arfrp1 or μ1-adaptin interact with Frizzled-6 or Celsr1, we performed GST-pull down analysis as before. GST-Arfrp1 and GST-μ1 bound HA-Vangl2 but not GFP-Frizzled-6 or GFP-Celsr1 in cell lysates from COS7 cells co-transfected with HA-Vangl2 and GFP-Celsr1 (*Figure 7J*) or co-transfected with HA-Vangl2 and GFP-Frizzled 6 (*Figure 7K*). These results suggest that sorting of Frizzled 6 and Celsr1 at the TGN is independent of the Arfrp1/AP-1 machinery.

## Arfrp1 regulates TGN export of protein tyrosine kinase 7

In addition to Vangl2, Arfrp1 is known to regulate TGN-to-plasma membrane trafficking of VSVG and E-cadherin (*Shin et al., 2005*; *Zahn et al., 2008*; *Nishimoto-Morita et al., 2009*). Each of these cargo molecules contains a basolateral sorting motif in the C-terminal cytosolic domain. Sequence alignment of protein tyrosine kinase 7 (PTK7), another plasma-membrane localized regulator of planar cell polarity (*Lu et al., 2004*), revealed a conserved tyrosine sorting motif (YVDL) in its predicted cytosolic domain (*Figure 8A*). We used a C-terminal Myc-His-tagged PTK7 (PTK7-Myc-His) to examine the effect of Arfrp1 depletion on traffic from the TGN. COS7 cells were transfected with control siRNA or siRNA against Arfrp1 and re-transfected after 48 hr with plasmids encoding PTK7-Myc-His. These conditions achieved an siRNA-specific depletion of Arfrp1 (*Figure 8H*). At steady state, around 50% of cells showed both surface- and Golgi-localized PTK7 in control cells and this localization pattern was not significantly changed in Arfrp1 knockdown cells. Given the high background of PTK7 delayed in the TGN in transfected COS7 cells, we adjusted the experimental conditions using a 20°C incubation followed by cycloheximide to synchronize a pool of newly-synthesized PTK7 in the TGN in control cells and Arfrp1 knockdown cells. After incubation at 20°C, a majority of cells (80%) showed strong accumulation of PTK7 at the TGN. After cells were returned to 32°C, a significantly higher percentage accumulated PTK7 at the TGN when Arfrp1 was depleted than in cells treated with control siRNA (*Figure 8B–G* and *Figure 8I*, 12 ± 8% vs 49 ± 6%). In contrast, the TGN localization of HA-Frizzled 6

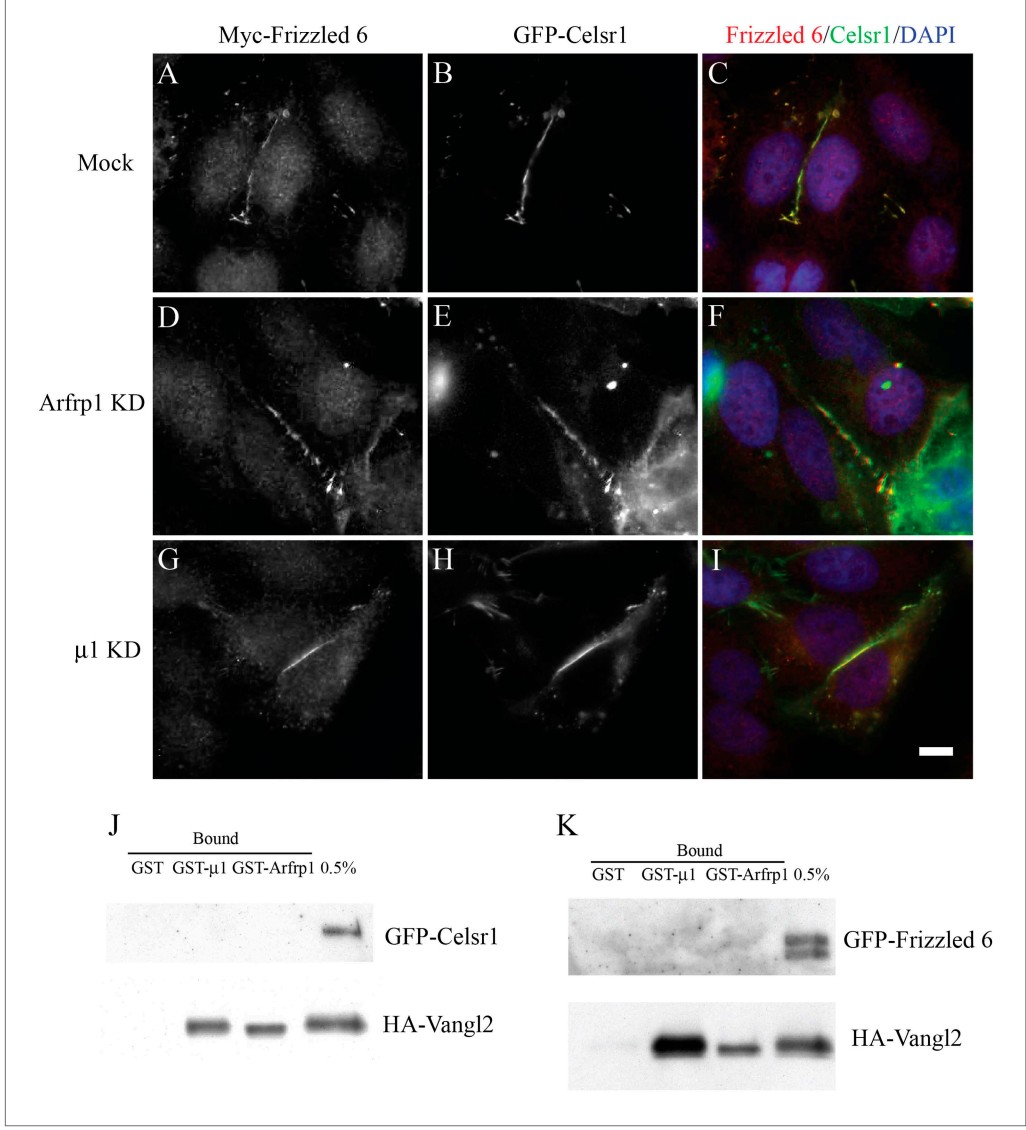

**Figure 7**. TGN export of Frizzled-6 and Celsr1 is independent of the Arfrp1/AP-1 machinery. (**A**)–(**I**). HeLa cells were either mock transfected (**A**–**C**) or transfected with siRNA against Arfrp1 (**D**–**F**) or μ1-adaptin (**G**–**I**) and re-transfected after 48 hr with plasmids encoding GFP-Celsr1 and Myc-Frizzled 6. After an additional 24 hr, cells were analyzed by immunofluorescence. Size bar = 10 μm. (**J**),(**K**). Cell lysates (250 μl) containing 1 mg/ml proteins from COS7 cells co-transfected with HA-Vangl2 and GFP-Celsr1 (**J**) or HA-Vangl2 and GFP-Frizzled 6 (**K**) were incubated with glutathione beads bearing 1 μg of GST, GTPγS-loaded GST-Arfrp1 or GST-μ1. The entire sample of bound HA-Vangl2, GFP-Celsr1 or GFP-Frizzled 6 were detected by immunoblot.

was not enhanced by depletion of Arfrp1 (*Figure 8J*). These results suggest that Arfrp1 also regulates TGN export of PTK7.

## Vangl2 and Frizzled 6 require protein kinase D for transport from the TGN

Protein kinase D (PKD) mediates membrane fission to generate TGN to cell surface transport carriers containing basolateral cargo molecules (*Yeaman et al., 2004*; *Malhotra and Campelo, 2011*). Expression of the kinase dead form of glutathione S-transferase tagged PKD2 (GST-PKD2-KD) or PKD3 (GST-PKD3-KD) in COS7 cells resulted in the accumulation of HA-Vangl2 and HA-Frizzled 6 at the juxtanuclear area, colocalized with the TGN marker, TGN46 (*Figure 9*). Thus, although Vangl2 and Frizzled 6 display distinct requirements for Arfrp1 and AP-1, they both depend on PKD for traffic from the TGN. We suggest that Vangl2 (and PTK7) and Frizzled 6 are sorted by independent means into

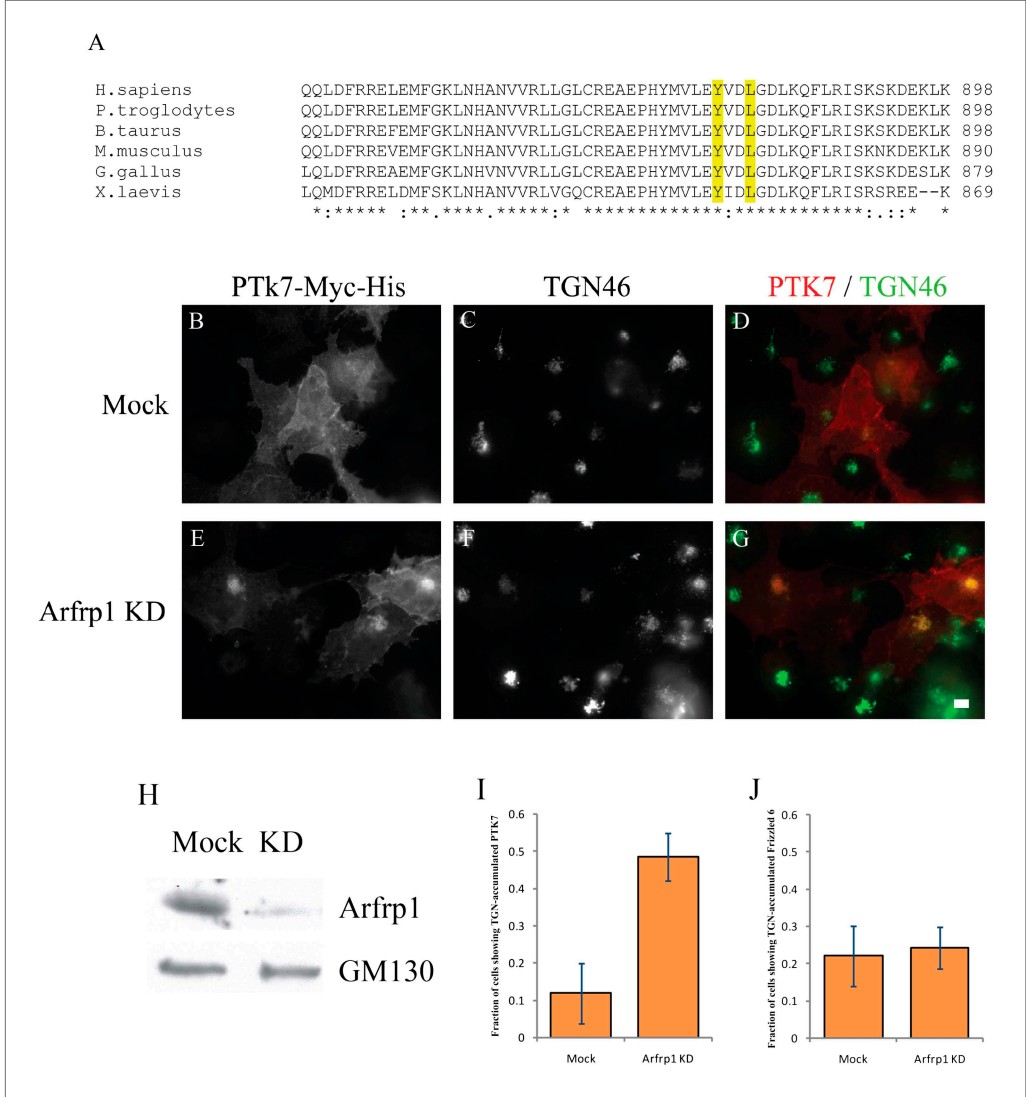

**Figure 8**. Arfrp1 regulates TGN export of PTK7. (**A**) Sequence alignment of PTK7 from different species reveals a conserved tyrosine sorting motif in its predicted C-terminal cytosolic domain. (**B**)–(**G**) COS7 cells were transfected with control siRNA or siRNA against Arfrp1 and re-transfected after 48 hr with plasmids encoding PTK7-Myc-His. After an additional 24 hr, cells were incubated at 20°C in the presence of 30 μg/ml cyclohexmide for 4 hr then shifted to 32°C for 90 min. After incubation, cells were analyzed by immunofluorescence using antibodies against His and TGN46. Size bar = 10 μm. (**H**) COS7 cell lysates from cells transfected with control siRNA or siRNA against Arfrp1 were analyzed by immunoblotting with anti-Arfrp1 antibody and, as a loading control, anti-GM130 antibody. (**I**) The fraction of cells showing TGN-accumulated PTK7 was quantified after incubation at 32°C (mean ± SD; N = 3; over 150 cells were counted for each group). (**J**) Similar siRNA knockdown and temperature shift experiments were performed in COS7 cells transfected with HA-Frizzled 6. The appearance of TGN-accumulated HA-Frizzled 6 was quantified in cells treated with control siRNA or siRNA against Arfrp1 after an incubation at 32°C (mean ± SD; N = 2; over 100 cells were counted for each group).

separate transport vesicles but that they share a common mechanism for membrane fission to form these carriers.

## Discussion

The TGN is an essential sorting station where newly-synthesized cargo proteins and lipids are packaged for transport to various destinations at the cell surface, extracellular matrix and the endosome system. The variety of cargo molecules and the need to reach diverse destinations has complicated the

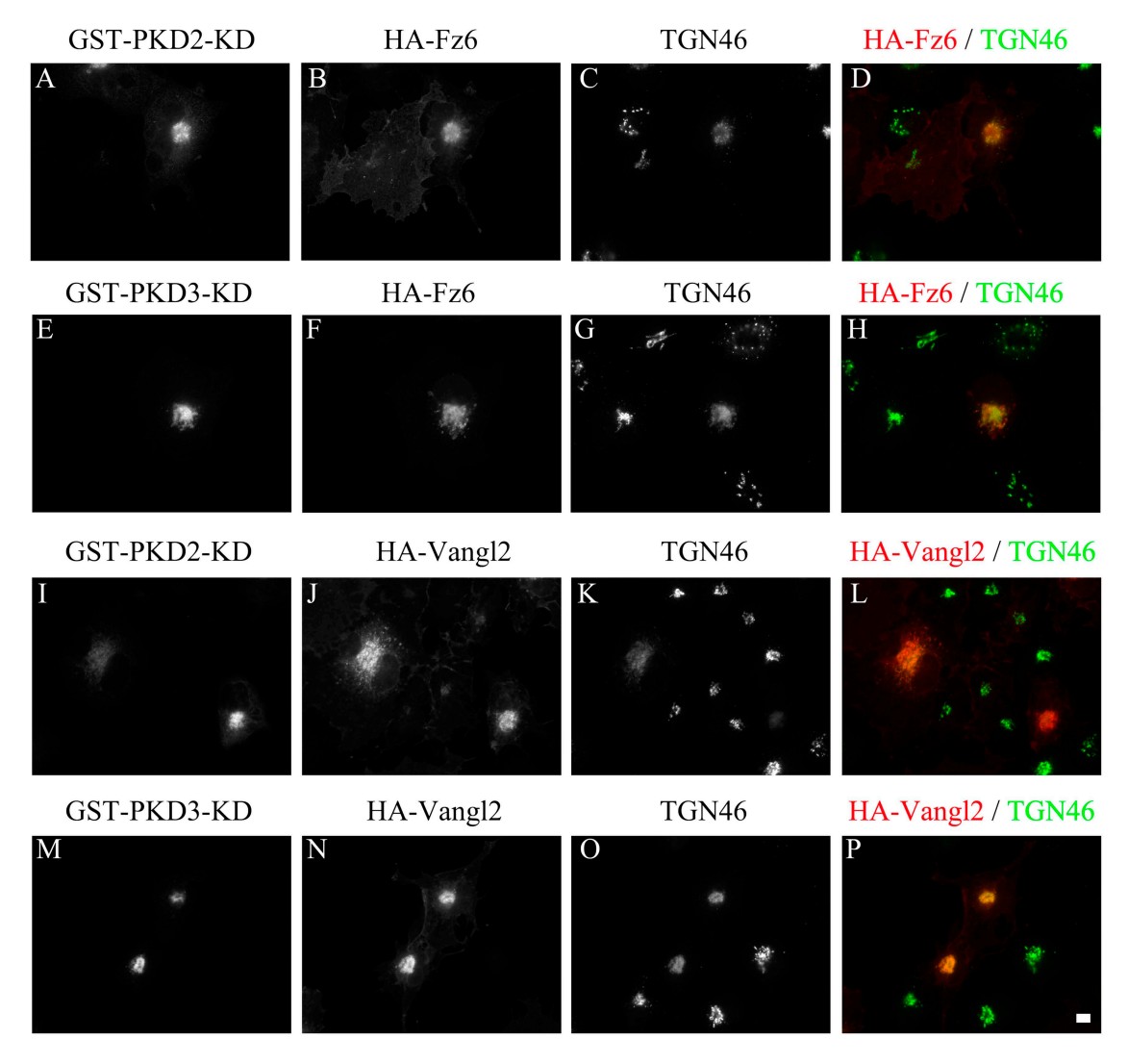

**Figure 9**. TGN export of Vangl2 and Frizzled 6 is protein kinase D dependent. COS7 cells were co-transfected with GST-PKD2-KD and HA-Frizzled 6 (**A**)–(**D**), GST-PKD3-KD and HA-Frizzled 6 (**E**)–(**H**), GST-PKD2-KD and HA-Vangl2 (**I**)–(**L**) or GST-PKD3-KD and HA-Vangl2 (**M**)–(**P**). Day 1 after transfection, cells were analyzed by immunofluorescence using anti-HA, anti-TGN46 and anti-GST antibodies. Size bar = 10 µm.

assignment of a general sorting mechanism from the TGN. At least some cargo traffic from the TGN depends on vesicle coat proteins that bind distinct sorting peptide motifs on the cytosolic domain of membrane cargo proteins (*De Matteis and Luini, 2008*; *Barfield et al., 2009*).

Here we show that TGN export of Vangl2, a PCP signaling receptor, depends on an unexpected complex of a TGN-localized Arf GTP-binding protein, Arfrp1, and the Golgi-localized clathrin adaptor complex, AP-1. siRNA knockdown of Arfrp1 or of subunits of AP-1 arrest Vangl2 traffic at the TGN as determined by co-localization of Vangl2 and the TGN marker, Golgin 97. Further, we have identified a sorting signal within the C-terminal cytoplasmic domain of Vangl2, YYXXF, the Phe residue of which is crucial for Vangl2 binding to AP-1 and traffic from the TGN to the cell surface. We propose a model wherein the interaction of Arfrp1-GTP and AP-1 exposes a sorting recognition determinant of the AP-1 µ subunit that binds the sorting motif on Vangl2 (*Figure 10A*), and this binding in turn helps to stabilize AP-1 assembly on membranes.

Mammalian cells possess two-dozen different Arf and Arf-like (Arl) proteins, only a few of which have been implicated in protein sorting or vesicle traffic (*Gillingham and Munro, 2007*). For example, Sar1 initiates COPII coat assembly at the ER (*Zanetti et al., 2012*); Arf1 is required for COPI-mediated

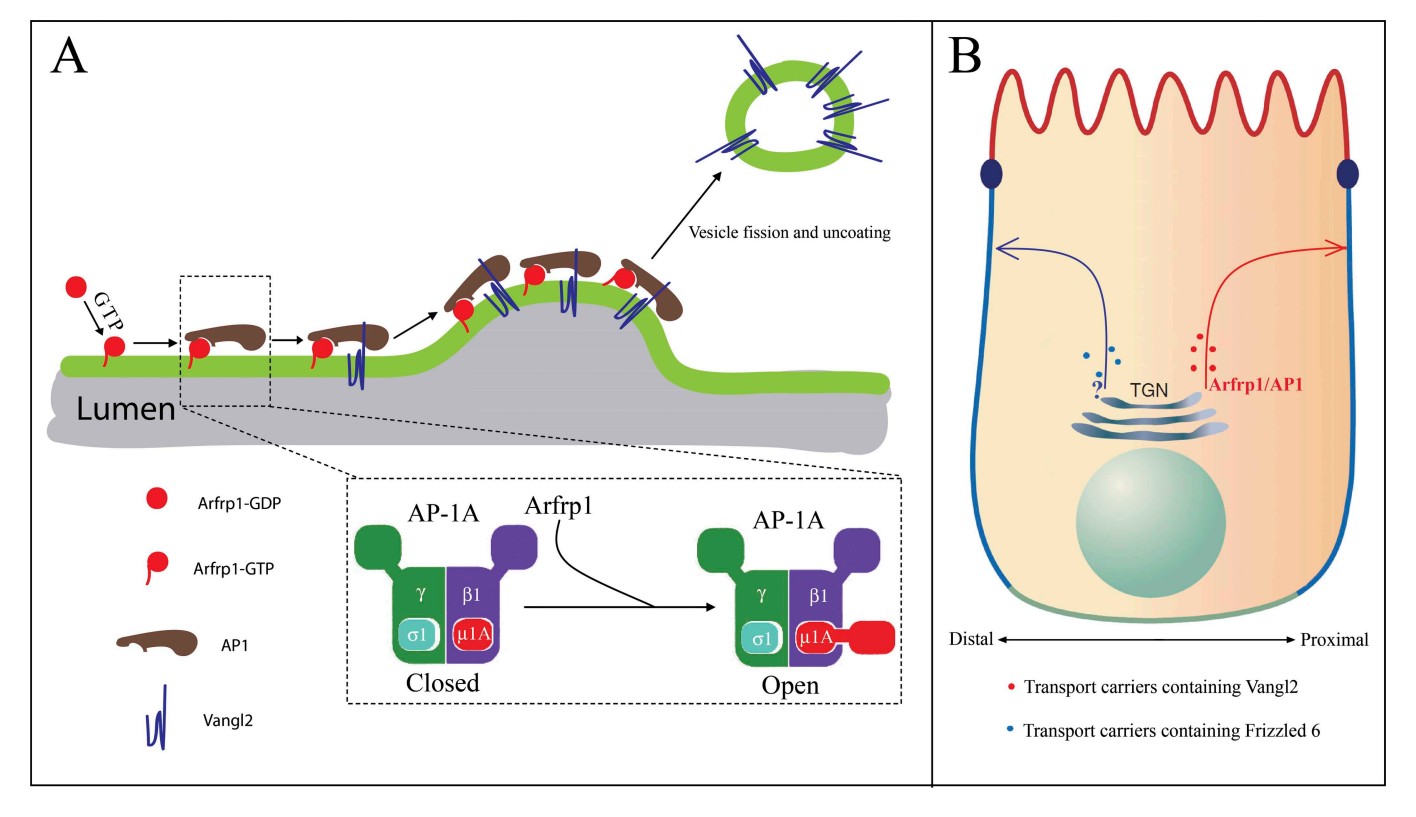

**Figure 10**. Proposed model. (**A**) Model depicting Arfrp1- and AP-1-mediated TGN sorting of Vangl2. Arfrp1 is recruited to TGN membranes upon GTP binding, possibly mediated by a TGN localized GEF. Subsequently, GTP-bound Arfrp1 recruits AP-1 to TGN membranes. GTP-bound Arfrp1 also promotes an open conformation of AP-1 that directly interacts with the tyrosine motif on Vangl2 cytosolic domain, thereby enriching Vangl2 in budding vesicles. Binding of Vangl2 cytosolic domain to AP-1, in turn, stabilizes the membrane association of AP-1 to allow sufficient time for AP-1 polymerization (possibly with clathrin as a coat outer layer) and vesicle budding. This model is consistent with reports showing that tyrosine sorting motifs promote membrane recruitment of AP-1 mediated by Arf1 (*Crottet et al., 2002*; *Lee et al., 2008*). (**B**) The asymmetrically localized PCP signaling molecules, including Vangl2 and Frizzled 6, are sorted by different sorting machineries for export from the TGN. Differential TGN sorting and polarized trafficking of these signaling receptors may contribute to their asymmetric distribution and the laterally polarized organization of epithelial cells.

vesicle budding and for the recruitment of clathrin AP-1 to the TGN and endosomes (*Gillingham and Munro, 2007*); Arl6 binds the BBsome to segregate cell surface proteins into the membrane of the primary cilium (*Jin et al., 2010*). Several of the Arfs and Arls are localized to the TGN of mammalian cells, and at least one, Arfrp1, is required for TGN to cell surface traffic of E-cadherin and VSV-G (*Shin et al., 2005*; *Zahn et al., 2008*; *Nishimoto-Morita et al., 2009*). Here we show that knockdown of Arfrp1 arrests the traffic of Vangl2 in a compartment that colocalizes with TGN markers but not with endosomal markers, confirming that Arfrp1 regulates trafficking from the late Golgi cisternae. Arfrp1 is essential at an early stage in mouse embryonic development (*Mueller et al., 2002*), possibly because it plays a role in the traffic of crucial cell surface proteins. However, unlike Sar1, which is required for traffic of all secretory cargo from the ER, Arfrp1 is not generally required for the transport of plasma membrane proteins from the TGN. For example, we show in this report that Frizzled 6, another PCP signaling receptor that is displayed on the distal cell surface opposite to Vangl2 on the proximal surface of epithelial cells, does not depend on Arfrp1 for its transit from the TGN.

Arfrp1 is proposed to regulate the membrane recruitment of Arl1 which in turn recruits GRIP domain-containing proteins to the TGN membrane (*Panic et al., 2003*; *Zahn et al., 2006*). However, at least one group reported that knockdown of Arfrp1 does not affect the localization of Arl1 and GRIP-domain containing proteins in mammalian cells (*Nishimoto-Morita et al., 2009*). Moreover, Arfrp1 and Arl1 appear to play different roles in trafficking between TGN and endosomes (*Nishimoto-Morita et al., 2009*). Our analysis indicates that knockdown of Arl1 does not affect the localization of

Vangl2 suggesting that Arl1 and its associated GRIP-domain containing proteins are not involved in TGN sorting of Vangl2.

Using immobilized GDP- and GTP-mutant forms of Arfrp1, we observed a GTP-selective interaction with the AP-1 complex in a crude cytosol fraction. Neither AP-2 nor AP-3 was detected in the proteins that bound to Arfrp1-GTP. We found that the μ and γ subunits of the AP-1 complex interact with Arfrp1 and that μ and γ subunits are required for the transit of Vangl2 from the TGN. Further, we observed that the residues of the YYXXF sorting motif required for traffic of Vangl2 are also crucial for the interaction of μ1-adaptin with Vangl2. The YYXXF sorting motif fits the consensus sequence of the canonical YXXΦ motif which has been identified in plasma membrane proteins that traffic to the basolateral surface in polarized cells. PTK7, another regulator of planar cell polarity also contains a conserved YXXΦ motif in its predicted C-terminal cytosolic domain and TGN export of PTK7 is also regulated by Arfrp1. Mutation in the canonical YXXΦ motif causes mis-sorting of basolateral proteins to the apical domains (*Muth and Caplan, 2003*). Here we show that alanine substitution of both of the tyrosine residues or alanine substitution of the phenylalanine residue completely blocks TGN export of Vangl2 to a greater extent than when Arfrp1 is depleted. The surface-localized Vangl2 may be retained during the course of the Arfrp1 knockdown whereas YXXΦ mutant Vangl2 may not reach the cell surface during the course of the transfection. Alternatively, a partially redundant Arf or Arl may replace Arfrp1 to mediate inefficient traffic of Vangl2 from the TGN.

Five different AP adaptor complexes have been identified in mammals, each serving a distinct role in traffic at the TGN, endosomes and cell surface (*Bonifacino and Lippincott-Schwartz, 2003*; *Hirst et al., 2011*). The μ subunit of each adaptor complex preferentially binds distinct but overlapping sets of YXXΦ motifs based on the identity of X and Φ residues and the residues surrounding the tyrosine sorting motif (*Ohno et al., 1998*). AP-1 regulates trafficking of mannose-6-P receptor from the TGN to endosomes (*Bonifacino and Lippincott-Schwartz, 2003*) and mediates TGN export of potassium channels (*Ma et al., 2011*). Given its central role in membrane traffic, deletion of various subunits of AP-1 leads to an embryonic lethal phenotype in the mouse (*Ohno, 2006*). In epithelial cells, some biosynthetic proteins traverse recycling endosomes en route to the basolateral membrane (*Fölsch et al., 2009*). Correspondingly, epithelial cells possess two isoforms of AP-1, a Golgi-localized AP-1A and a recycling endosome-localized AP-1B. AP-1A is proposed to mediate TGN export, thus this isoform may participate in Vangl2 traffic. We have not explored the post-Golgi pathway Vangl2 takes en route to the cell surface, thus it remains possible that Vangl2 invokes a recycling endosome in its itinerary to the proximal surface of an epithelial cell.

Membrane recruitment of AP-1 is proposed to require Arfs and PI4P (*Wang et al., 2003*). Another adaptor-like protein, GGA, has been shown to mediate membrane recruitment of PI4-kinase, which may then create a binding site for AP-1 (*Daboussi et al., 2012*). We find that the sorting motif in the C-terminal domain of Vangl2 enhances AP-1 binding to Arfrp1-GTPγS on the surface of synthetic liposomes. Similarly, Arf1-GTP binding to AP-1 is promoted by a peptide containing the sorting signal on the cation-independent mannose 6-phosphate receptor (*Lee et al., 2008*) and recruitment of AP-1 to synthetic liposomes requires tyrosine sorting motifs (*Crottet et al., 2002*). Structural analysis has suggested that adaptor complexes have open and closed cargo binding sites whose transition is implicated to be influenced by Arf-GTP binding (*Figure 10A*) (*Jackson et al., 2010*; *Yu et al., 2012*). Binding of a cargo-sorting motif to the open state may then stabilize coat assembly on membranes in preparation for transport vesicle budding.

Our results build on this model of cargo capture to suggest that coat-adaptors may have more than two active conformations influenced by different Arf proteins. In the case of Vangl2, we propose that Arfrp1 and the Vangl2 sorting motif favor an open conformation-exposed μ subunit of AP-1 that is not available in the complex of Arf1 and AP-1. This combination may be responsible for the capture of cargo proteins destined for transport to the proximal cell surface domain in polarized epithelial cells. In a distinct example, a YKFFE sequence recognized by AP-4 directs the traffic of amyloid precursor protein (APP) from the TGN to early endosomes (*Burgos et al., 2010*). This motif binds to a novel site on the surface of the μ4 subunit opposite the canonical tyrosine-signal-binding site. Key residues in this novel binding site are conserved in the μ subunits of other AP complexes (*Burgos et al., 2010*). The YYXXF motif on Vangl2 could occupy an alternative site on μ1, and Arfrp1-AP-1, but not Arf1-AP-1, may promote the exposure of this site. Currently, there is no direct structural evidence for this possibility. In contrast to these examples, Frizzled, which appears to be transported independent of Arfrp1 and AP-1, may rely on another Arf and adaptor protein for traffic to the distal cell surface domain (*Figure 10B*).

# Materials and methods

## Constructs and reagents

Small interference siRNAs against Arfrp1 or against subunits of different adaptor complexes were purchased from Qiagen (Valencia, CA), Thermo Scientific (Rockford, IL) or Ambion (Grand Island, NY). The target sequence against Arfrp1 was CACCACCACCGTGGGCCTAAA. The target sequence against μ1-adaptin was AAGGCATCAAGTATCGGAAGA. The target sequence against γ1-adaptin was TAGCACAGGTTGCCACTAA. The target sequence against δ3-adaptin was CGCTGAAAATTCCTATGTT. The target sequence against μ3-adaptin was CCAAGGTACTAACATGGGA. Antibodies and dilutions for immunoblotting were: mouse anti-μ1a (Abnova, Taipei, Taiwan, 1:2000), mouse anti-arfrp1 (Abnova, 1:500), mouse anti-γ1 (BD Transduction Laboratory, San Jose, CA, 1:2000), rabbit anti-μ3 (Proteintech Group, Chicago, IL, 1:2000), mouse anti-δ3 (Rockland, Gilbertsville, PA, 1:2000), mouse anti-Golgin 97 (Invitrogen, Grand Island, NY, 1:500 for immunofluorescence (IF)), rabbit anti-MBP (New England Biolabs, Ipswich, MA, 1:4000), mouse anti-GM130 (BD Biosciences, San Jose, CA, 1:500 for IF), rabbit anti-HA (Cell Signaling, Danvers, MA, 1:200 for IF, 1:2000 for immunoblotting), mouse anti-GFP (Santa Cruz Biotechnology, Santa Cruz, CA, 1:2000), mouse anti-Myc (Cell Signaling, Danvers, MA, 1:2000), mouse anti-EEA1 (BD Biosciences, San Jose, CA, 1:2000), mouse anti-tubulin (Abcam, Cambridge, MA, 1:2500), rabbit anti-CRMP2 (antibody-online, Atlanta, GA, 1:3000), mouse anti-His (Qiagen, CA, 1:200 for IF and 1:2000 for immunoblotting), sheep anti-TGN46 (AbD Serotec, UK, 1:200), rabbit anti-Rab11 (Invitrogen, Grand Island, NY, 1:200), goat anti-Rab7 (Santa Cruz Biotechnology, CA, 1:200) and goat-anti-dynamin II (Affinity Bioreagent, Golden, CO, 1:2000).

## Cell culture, immunofluorescence, transfection and image analysis

HeLa cells, HeLa cells stably expressing HA-Vangl2 and COS7 cells were maintained in GIBCO Dulbecco's Modified Eagle Medium containing 10% Fetal Bovine Serum (FBS), 10 mU/mL of penicillin and 0.1 mg/mL of streptomycin. Transfection of siRNA or DNA constructs into HeLa cells or COS7 cells was performed using lipofectamine 2000 as described in the manual provided by Invitrogen. For immunofluorescence, cells growing on coverslips were fixed in 4% PFA for 20 min then washed five times with 500 μl of PBS and incubated with permeabilization buffer (PBS containing 0.1% TX-100, 0.2 M Glycine, 2.5% FBS) at RT for 30 min. Then cells were incubated with primary antibody and secondary antibody in permeabilization buffer for 30 min. Each antibody incubation was following by five times wash with PBS.

Images were acquired with a Zeiss LSM 510 confocal microscope system or a Zeiss Axioobserver Z1 microscope system. Image J (http://rsb.info.nih.gov/ij/) was used for colocalization analysis (*Guo et al., 2008*). Briefly, the two images were adjusted to be the same average intensity of pixel value using divide function. A threshold was chosen manually to select the area stained with a Golgi marker. Subsequently, the numbers of above threshold pixels were determined for each Golgi marker (A and B). Colocalized pixels were determined using the colocalization function with a fixed ratio of 0.75 (C). Finally the value of colocalization was determined by the average value of C/A and C/B.

## Temperature shift/cycloheximide experiment

COS7 cells were transfected with control siRNA or siRNA against Arfrp1 and re-transfected after 48 hr with plasmids encoding PTK7-Myc-His or HA-Frizzled 6. After an additional 24 hr, cells were incubated in opti-MEM (Invitrogen, Grand Island, NY) containing 10% FBS and 30 μg/ml cycloheximide at 20°C for 4 hr to accumulate cargo proteins at the TGN. Cells were then shifted to 32°C for 90 min to restore transport from the TGN and analyzed by immunofluorescence (*Wakana et al., 2012*).

## Protein purification

Glutathione transferase (GST) fusion protein purification was performed as described previously (*Guo et al., 2008*). Briefly, full length constructs for μ1, Arfrp1 wild type, and T31N and Q79L mutants were cloned in a pGEX-2T vector (GE Healthcare Biosciences, NJ). The constructs were transformed in BL21 cells and individual colonies were grown to O.D. 0.6 in 500 ml of Luria broth (LB) at 37°C. Protein expression was induced with 0.5 mM isopropyl-1-thio-β-d-galactopyranoside (IPTG) for 5 hr at 25°C. Cells were centrifuged, washed with PBS and lysed in lysis buffer (50 mM Tris, pH 8.0, 5 mM EDTA, 150 mM NaCl, 10% glycerol, 5 mM dithiothreitol, 0.5 mg/ml lysosome, protein-ase inhibitor cocktail, complete, EDTA free, one tablet for 50 ml solution, Roche, Mannheim, Germany). After 30 min on ice, the cell lysates were adjusted to contain 0.5% TX-100 and sonicated

four times for 30 s each time and centrifuged at 55k for 30 min in a Beckman TLA 100.3 rotor for the ultracentrifuge. The supernatant fraction was incubated with 250 µl glutathione-agarose beads for 2 hr at 4°C. After incubation, the beads were washed four times with PBS containing 1 mM DTT and 0.1% Tween 20 then two times with PBS. The beads were either used for a binding assay or mixed with elution buffer (50 mM Tris, pH 8.0, 250 mM KCl, 1 mM DTT, 25 mM glutathione, pH 8.0, proteinase inhibitor cocktail). MBP and His fusion protein purification were performed according to the protocol provided by Qiagen (Valencia, CA) or New England Biolabs (Ipswich, MA) respectively.

The AP-1 complex was purified as described previously (*Lee et al., 2008*). Cyanogen bromide-activated Sepharose-4B beads (6 mg, GE Healthcare Biosciences, NJ) were incubated with 1ml 1 mM HCl on ice for 15 min, then the beads were washed four times with 1 ml coupling buffer (0.1 M NaHCO$_3$ pH 8.3, 0.5 M NaCl) and incubated with 30 µg mouse antibody against γ1-adaptin (100/3, Abcam, MA) in 750 µl coupling buffer at 4°C overnight. After incubation, the beads were washed five times with 1 ml coupling buffer and then transferred to 0.1 M Tris–HCl buffer, pH 8.0, and incubated on ice for 2 hr followed by washing with at least three cycles of buffer at alternative pHs (coupling buffer followed by 0.1 M acetic acid/sodium acetate, pH 4.0, 0.5 M NaCl). Beads were then incubated with 3 ml 8 mg/ml bovine brain cytosol prepared as described by *Christoforidis and Zerial (2000)* in HKM buffer (20 mM Hepes, pH 7.4, 100 mM KCl, 5 mM MgCl$_2$) at 4°C overnight. After incubation, the beads were washed four times with 1 ml HKM buffer and then eluted with 150 µl HKM buffer containing 0.3 mg/ml peptide corresponding to the hinge region of γ1-adaptin at 4°C for 5 hr. The eluted fraction was dialyzed against HKM buffer.

## Binding assays

A modified protocol from Jin et al. was performed to detect proteins that bind specifically to the GTP-bound Arfrp1 (*Jin et al., 2010*). Briefly, GST fused to dominant negative or dominant active forms of Arfrp1 were purified from bacteria in a lysis buffer containing 5mM EDTA to extract Mg$^{2+}$ and nucleotides. Proteins (50 µg) on glutathione beads was incubated with nucleotide loading buffer (20 mM Hepes, pH 7.4, 100 mM KCl, 5 mM MgCl$_2$, 500 µM GDP or GTPγS, proteinase inhibitor cocktail) at room temperature for 1 hr. After incubation, the beads were mixed with bovine brain cytosol in binding buffer (20 mM Hepes, pH 7.4, 100 mM KCl, 5 mM MgCl$_2$, 100 mM GDP or GTPγS, 0.1% TX100, proteinase inhibitor cocktail) at 4°C overnight. Beads were then mixed with washing buffer (20 mM Hepes, pH 7.4, 500 mM KCl, 5 mM MgCl$_2$, 100 mM GDP or GTPγS, 0.1% TX100, proteinase inhibitor cocktail) and then with washing buffer without nucleotide and MgCl$_2$. Bound proteins were desorbed with elution buffer (20 mM Hepes, 500 mM KCl, 1 mM reversed GDP or GTPγS, 0.1% TX100, proteinase inhibitor cocktail, 25 mM EDTA). Eluted fractions were concentrated in Amicon ultracentrifuge filters, and samples were electrophoresed on a 4–20% gradient gel which was stained with a silver staining kit (Silver Quest, Invitrogen). Aliquots of the eluted fraction were also processed for immunoblot analysis.

Binding assays to detect interactions between µ1-adaptin and various Vangl2 constructs were carried out with 4 µl of compact glutathione beads bearing 1 µg of GST-µ1. The beads were incubated with 0.5 µg purified MBP-Vangl2 cytosolic domain wild type or mutant constructs in 400 µl binding buffer (100 mM KCl, 20 mM Hepes, pH 7.4, 5 mM MgCl$_2$, 0.5% TX-100) containing 0.1 mg/ml BSA, or incubated with 150 µl 0.2mg/ml cell lysates from COS7 cells transiently transfected with HA-Vangl2 wild type or mutant constructs, in binding buffer at 4°C for 90 min. After incubation, the beads were washed with four times with 500 µl binding buffer and the bound material was analyzed by immunoblot.

## Yeast two-hybrid assay

The yeast two-hybrid assay was carried out as described previously (*Ohno et al., 1998*). The yeast strain (PJ69-4A) was cotransformed with mouse µ1A construct in pACT2 and Vangl2 cytosolic domain wild type or mutant constructs in pGBT9. Colonies coexpressing both constructs were selected by their ability to grow on selective medium (dropout without tryptophan and leucine). After selection for 3 days, individual colonies were inoculated in selective medium at 30°C overnight. The colonies were then resuspended in water and the cell concentration was adjusted to OD$_{600}$ = 1.0 and serial dilutions were generated. Equal amount of cells for each serial dilution were spotted on selective medium and pictures were taken after 3 days of growth on the selective medium.

## Liposome flotation assay

Lipids and cholesterol, except Texas red PE, were purchased from Avanti (Alabaster, Alabama). Texas red PE was purchased from Invitrogen (Grand Island, NY). Lipids and cholesterol were mixed in chloroform in

the following molar ratio (*Bacia et al., 2011*): 51% 1,2, dioleoyl-sn-glycero-3-phosphocholine (DOPC), 22% 1,2, dioleoyl-sn-glycero-3-phosphoethanolamine (DOPE), 8% 1,2, dioleoyl-sn-glycero-3-[phospho-L-serine]sodium salt (DOPS), 5% 1,2, dioleoyl-sn-glycero-3-phosphate (monosodium salt) (DOPA), 8% L-α-phosphatidylinositol (PI), 2.2% L-α-phosphatidylinositol-4-phosphate (PI4P), 0.8% L-α-phosphatidylinositol-4,5-bisphosphate (PI(4,5)P2), 2% 1,2, dipalmitoyl-sn-glycero-3-(cytidine diphosphate) (CDP-DAG), 1% Texas red 1,2, dihexadecanoyl-sn-glycero-3-phosphoetanolamine (TX-PE) and cholesterol (20% by weight). Chloroform was evaporated in a vacuum with an argon flow and rotation in a 37°C water bath. Liposomes were generated by rotating the dried lipid film in HKM buffer (20 mM Hepes, pH 7.4, 100 mM KCl, 5 mM MgCl$_2$) in a 37°C water bath for 2 hr. Liposomes were extruded to achieve homogeneity in size using the Mini-Extruder (Avanti Polar Lipids, Inc.) and Nuclepore track-etched membranes with 400-nm pores (Whatman, Sanford, ME).

Samples containing 1.5 µg of Arfrp1-His, Arfrp1-FLAG or Arf1-FLAG in the presence or absence of 1 µg MBP-Vangl2 cytosolic domain wild type or tyrosine mutant were incubated with 8 µl of 1.8 mg/ml liposomes in HKM buffer containing 100 µM nucleotides at room temperature for 45 min in 50 µl of reaction mixture. After incubation, 2 µg purified AP-1 was added and incubated for an additional 1 hr at RT. The reaction mixture was adjusted to 1.75 M sucrose and overlayed with 100 µl 0.75 M sucrose and 30 µl HKM buffer. The samples were centrifuged at 55,000 rpm in a TLS55 rotor in the Beckman-ultracentrifuge for 2.5 hr at 4°C. Fractions were collected from the bottom of the tube using a peristaltic pump (RAININ, Columbus, OH) and aliquots were analyzed by SDS-PAGE and immunoblot. Proteins were visualized and quantified using a Bio-Red GelDot imaging system. Flotation of liposomes after centrifugation was monitored by following Texas Red-PE fluorescence.

## Acknowledgements

The authors thank Dr Vivek Malhotra (CRG, Spain) for kindly providing the constructs of GST-PKD2-KD and GST-PKD3-KD; Dr Juan S. Bonifacino (NIH) for kindly providing the constructs for Yeast-two-hybrid analysis; Dr Stuart Kornfeld (Washington University in St. Louis) for kindly providing regents for purify the AP-1 complex; Ann Fischer and Michelle Richner for tissue culture support; Devon Jensen for providing various constructs of PCP signaling receptors; Kanika Pahuja and Pengcheng Zhang for comments on the manuscript. Y.G is an Associate of the HHMI, G.Z. is an HFSP fellow and R.S. is an Investigator of the HHMI and a Senior Fellow of the Miller Institute, University of California, Berkeley.

## Additional information

### Competing interests

RS: Editor-in-Chief, *eLife.* The other authors have declared that no competing interests exist.

### Funding

| Funder | Grant reference number | Author |
|---|---|---|
| Howard Hughes Medical Institute | | Yusong Guo |
| Howard Hughes Medical Institute | | Randy Schekman |

The funder had no role in study design, data collection and interpretation, or the decision to submit the work for publication.

### Author contributions

YG, Conception and design, Acquisition of data, Analysis and interpretation of data, Drafting or revising the article; GZ, Conception and design, Drafting or revising the article; RS, Conception and design, Drafting or revising the article

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
