## [Decision Letter]

Thank you for choosing to send your work entitled “A Novel GTP-binding
protein-adaptor protein complex responsible for export of Vangl2 from the trans Golgi
network” for consideration at eLife. Your article has been evaluated by a Senior
Editor (Detlef Weigel) and 3 reviewers, one of whom is a member of our Board of
Reviewing Editors (Suzanne Pfeffer).

The Reviewing Editor and the other reviewers discussed their comments before we reached
this decision, and the Reviewing Editor has assembled the following comments based on
the reviewers' reports. Our goal is to provide the essential revision requirements
as a single set of instructions, so that you have a clear view of the revisions that are
necessary for us to publish your work.

Overall, this is a well-done study that provides convincing evidence that
Arfrp1·GTP, in conjunction with AP-1, is required for the transport of the
transmembrane protein Vangl2 to the cell surface. This new finding is of particular
interest in that Arf1 cannot replace Arfrp1 in this function. The experiments are well
controlled, the data are clean, and the text is very clearly written. We would be
pleased to publish the work in eLife. The reviewers included true experts in this field
who evaluated the work with great care. It should be straightforward to respond
textually to each of the following minor comments in a revised manuscript.

1. It is unclear from the data how much of the TGN deposited AP-1 is Arf1 dependent and
how much is Arfrp1 dependent. Arfrp1 is reported to be relatively Brefeldin A resistant
compared with Arf1. Do you know how much AP-1 remains after BFA application?

2. Please explain why such a different experimental strategy was required for evaluating
PTK7 when for the other transmembrane cargoes, simple RNAi was sufficient to see a
steady-state morphologic change. Also, the conclusion that Frizzled 6 and Celsr1 were
not dependent on Arfrp1 or AP-1 is not too surprising, as neither has AP-1-recognized
sorting signals. Therefore, please tone down the statement that PTK7 “depends on
Arfrp1 for traffic”. What would Frizzled 6/Celsr1 localization look like after a
similar temperature shift/cycloheximide experiment?

3. Please mention the possible role of Arl1 or Golgin245 in the phenotypes observed.
Others have reported these to be downstream targets of Arfrp1.

4. The Discussion section should be enhanced and could include the following points. 1.
How a YXXphi-like sequence can be selectively recognized by only one of the
YXXphi-decoding APs. Why wouldn't AP-2 and AP-3 recognize the YYxxphi signal? Is it
deactivated by phosphorylation post TGN perhaps? 2. At the end, you suggest that
Arfrp1-GTP directs AP-1 to a conformation that allows Vangl2 engagement, a conformation
not generated by Arf1-GTP binding. Is there structural precedent for this? What is the
degree of amino acid conservation between these two GTPases? 3. Why is there evidently a
complete block of Vangl2 delivery when the YYxxF sequence is mutated but only a partial
block with Arfrp1 knockdown? 4. Why do Arfrp1 and Arf1 bind differentially to allow this
cooperativity with the Vangl2 Yxxphi? Please consider the key role of the distal Phe vs.
the proximal Tyr pair in the Vangl2 YYxxF signal. Could this be binding to the opposite,
convex surface of the AP-1 mu1 subunit, similar to the Phe-based APP signal binding to
AP-4 mu4?

---

## [Author Response]

We thank the Editor and reviewers for providing positive and valuable comments.

*1. It is unclear from the data how much of the TGN deposited AP-1 is Arf1
dependent and how much is Arfrp1 dependent. Arfrp1 is reported to be relatively
Brefeldin A resistant compared with Arf1. Do you know how much AP-1 remains after BFA
application*?

We analyzed the localization of γ1-adaptin in BFA treated COS7 cells. We found
that the majority of γ1-adaptin was dispersed from the TGN after 15min of BFA
treatment, which is consistent with previous reports (Shin et al., 2005). It is possible
that most of the AP-1 is bound to BFA sensitive Arfs such as Arf1, and only a small
fraction is bound to Arfrp1, which is relatively BFA resistant. BFA could also cause
indirect effects on structure and lipid composition of the TGN, which may affect AP-1
recruitment.

*2. Please explain why such a different experimental strategy was required for
evaluating PTK7 when for the other transmembrane cargoes, simple RNAi was sufficient
to see a steady-state morphologic change. Also, the conclusion that Frizzled 6 and
Celsr1 were not dependent on Arfrp1 or AP-1 is not too surprising, as neither has
AP-1-recognized sorting signals. Therefore, please tone down the statement that PTK7
“depends on Arfrp1 for traffic”. What would Frizzled 6/Celsr1
localization look like after a similar temperature shift/cycloheximide
experiment*?

We didn't detect an obvious difference in PTK7 localization in Arfrp1 knockdown and
control cells at steady state. A significant percentage (around 50%) of cells showed
both surface- and Golgi-localized PTK7 at steady state in cells treated with control
siRNA. After the temperature shift/cycloheximide experiment, the percentage of control
cells showing Golgi-localized PTK7 dropped to around 12%. Thus we adopted this regimen
to synchronize PTK7 in the Golgi region to localize the effect of Arfrp1 knockdown
specifically on exit from the Golgi. This is now described in the manuscript. We agree
that the defects of PTK7 trafficking in Arfrp1 knockdown cells are likely due to a
kinetic delay. Accordingly, we have modified the statement that PTK7 “depends on
Arfrp1 for traffic”. We analyzed the localization of Frizzle 6 in the Arfrp1
knockdown cells in a similar temperature shift/cycloheximide experiment and did not see
a difference (now shown in the text and Figure 8J).

*3. Please mention the possible role of Arl1 or Golgin245 in the phenotypes
observed. Others have reported these to be downstream targets of Arfrp1*.

In yeast, membrane recruitment of Arl1 and GRIP domain containing Golgins is reported to
be mediated by Arl3, the yeast equivalent of Arfrp1 (Panic et al., 2003). However, in
mammalian cells, there are contradictory reports regarding the role of Arfrp1 in
regulating membrane recruitment of Arl1 and Golgin245 (Nishimoto-Morita et al., 2009;
Zahn et al., 2006). Our siRNA knockdown screen indicates that knockdown of Arl1 does not
affect the localization of Vangl2, suggesting that Arl1 and its associated GRIP-domain
containing proteins are not involved in TGN sorting of Vangl2 (this is now discussed in
the manuscript).

*4. The Discussion section should be enhanced and could include the following
points. 1. How a YXXphi-like sequence can be selectively recognized by only one of
the YXXphi-decoding APs. Why wouldn't AP-2 and AP-3 recognize the YYxxphi
signal? Is it deactivated by phosphorylation post TGN perhaps? 2. At the end, you
suggest that Arfrp1-GTP directs AP-1 to a conformation that allows Vangl2 engagement,
a conformation not generated by Arf1-GTP binding. Is there structural precedent for
this? What is the degree of amino acid conservation between these two GTPases? 3. Why
is there evidently a complete block of Vangl2 delivery when the YYxxF sequence is
mutated but only a partial block with Arfrp1 knockdown? 4. Why do Arfrp1 and Arf1
bind differentially to allow this cooperativity with the Vangl2 Yxxphi? Please
consider the key role of the distal Phe vs. the proximal Tyr pair in the Vangl2 YYxxF
signal. Could this be binding to the opposite, convex surface of the AP-1 mu1
subunit, similar to the Phe-based APP signal binding to AP-4 mu4*?

4.1. The μ subunit of each adaptor complex preferentially binds distinct but a
considerable degree of overlapping sets of YXXØ motifs (Ohno et al., 1998; Ohno et
al., 1996). The affinity of YXXØ motifs to different adaptor medium subunits is
regulated by the identity of the X and Ø residues and the residues surrounding the
tyrosine sorting motif as well as the position of the motif within the cytosolic domain
(Ohno et al., 1998; Ohno et al., 1996). This is now discussed in the text. Vangl2 is
phosphorylated at the cell surface and the N terminal- but not C terminal-cytosolic
domain is the major phosphorylation region (Gao et al., 2011). It would be interesting
to test whether the tyrosine residues in the YYXXF motif on Vangl2 are phosphorylated to
deactivate the interaction between AP-1 and Vangl2, but this is outside of the scope of
our current study.

4.2. Currently, there is no direct structural evidence suggests that AP-1 has multiple
alternative conformations to accommodate different Arfs and sorting motifs. Arfrp1
shares 34% sequence identity with Arf1. This is now described in the manuscript.

4.3. Two possible reasons could account for this. One is that the plasma-membrane pool
of Vangl2 may be synthesized and transported to the plasma membrane before the siRNA
treatment depletes Arfrp1. Experiments with mutant Vangl2 may not be subject to this
limitation. Alternatively, another Arf or Arl protein may engage AP-1 and the YYXXF
motif at a low level of efficiency to transport Vangl2 out of the trans-Golgi. This is
now discussed in the manuscript.

4.4. One can speculate on alternative explanations, but we favor the idea that AP-1 may
assume several different conformations to accommodate members of the Arf/Arl GTPases and
sorting motifs. The key residues in the novel cargo binding site on μ4 are
conserved in the μ subunits of other AP complexes (Burgos et al., 2010). Thus it
is possible that μ1 could have multiple cargo binding sites that recognize
specific tyrosine sorting motifs. We speculate that the YYXXF motif on Vangl2 could bind
a novel cargo-binding site on μ1; Arfrp1, but not Arf1, bound to AP-1 could
promote the exposure of this novel cargo-binding site on μ1-adaptin. This is now
described in the manuscript.

**References**

Burgos, P.V., Mardones, G.A., Rojas, A.L., daSilva, L.L., Prabhu, Y., Hurley, J.H., and
Bonifacino, J.S. (2010). Sorting of the Alzheimer's disease amyloid precursor
protein mediated by the AP-4 complex. Developmental cell *18,*
425-436. 

Gao, B., Song, H., Bishop, K., Elliot, G., Garrett, L., English, M.A., Andre, P.,
Robinson, J., Sood, R., Minami, Y., *et al.* (2011). Wnt signaling
gradients establish planar cell polarity by inducing Vangl2 phosphorylation through
Ror2. Developmental cell *20,* 163-176. 

Nishimoto-Morita, K., Shin, H.W., Mitsuhashi, H., Kitamura, M., Zhang, Q., Johannes, L.,
and Nakayama, K. (2009). Differential effects of depletion of ARL1 and ARFRP1 on
membrane trafficking between the trans-Golgi network and endosomes. The Journal of
biological chemistry *284,* 10583-10592. 

Ohno, H., Aguilar, R.C., Yeh, D., Taura, D., Saito, T., and Bonifacino, J.S. (1998). The
medium subunits of adaptor complexes recognize distinct but overlapping sets of
tyrosine-based sorting signals. The Journal of biological chemistry
*273,* 25915-25921. 

Ohno, H., Fournier, M.C., Poy, G., and Bonifacino, J.S. (1996). Structural determinants
of interaction of tyrosine-based sorting signals with the adaptor medium chains. The
Journal of biological chemistry *271,* 29009-29015. 

Zahn, C., Hommel, A., Lu, L., Hong, W., Walther, D.J., Florian, S., Joost, H.G., and
Schurmann, A. (2006). Knockout of Arfrp1 leads to disruption of ARF-like1 (ARL1)
targeting to the trans- Golgi in mouse embryos and HeLa cells. Molecular membrane
biology *23,* 475-485.